# Antimicrobial Peptides Display Strong Synergy with Vancomycin Against Vancomycin-Resistant *E. faecium*, *S. aureus*, and Wild-Type *E. coli*

**DOI:** 10.3390/ijms21134578

**Published:** 2020-06-27

**Authors:** Chih-Lung Wu, Ju-Yun Hsueh, Bak-Sau Yip, Ya-Han Chih, Kuang-Li Peng, Jya-Wei Cheng

**Affiliations:** 1Institute of Biotechnology and Department of Medical Science, National Tsing Hua University, Hsinchu 300, Taiwan; ericwu.cl@gmail.com (C.-L.W.); sandvik878@gmail.com (J.-Y.H.); neuron.hch@gmail.com (B.-S.Y.); chih80517@gmail.com (Y.-H.C.); richard850210@gmail.com (K.-L.P.); 2Department of Neurology, National Taiwan University Hospital Hsinchu Branch, Hsinchu 300, Taiwan

**Keywords:** antimicrobial peptide, antibiotic resistance, vancomycin, synergism, bulky non-nature amino acid

## Abstract

There is an urgent and imminent need to develop new antimicrobials to fight against antibiotic-resistant bacterial and fungal strains. In this study, a checkerboard method was used to evaluate the synergistic effects of the antimicrobial peptide P-113 and its bulky non-nature amino acid substituted derivatives with vancomycin against vancomycin-resistant *Enterococcus faecium, Staphylococcus aureus*, and wild-type *Escherichia coli*. Boron-dipyrro-methene (BODIPY) labeled vancomycin was used to characterize the interactions between the peptides, vancomycin, and bacterial strains. Moreover, neutralization of antibiotic-induced releasing of lipopolysaccharide (LPS) from *E. coli* by the peptides was obtained. Among these peptides, Bip-P-113 demonstrated the best minimal inhibitory concentrations (MICs), antibiotics synergism, bacterial membrane permeabilization, and supernatant LPS neutralizing activities against the bacteria studied. These results could help in developing antimicrobial peptides that have synergistic activity with large size glycopeptides such as vancomycin in therapeutic applications.

## 1. Introduction

The emergence of antibiotic-resistant bacterial and fungal strains has become a serious public health problem worldwide [1]. Vancomycin has proved to be effective in treating a number of infections caused by multidrug-resistant Gram-positive pathogens such as methicillin-resistant *Staphylococcus aureus* (MRSA). However, strains with reduced susceptibility to vancomycin are known to occur [2]. Moreover, for Gram-negative bacteria, their outer membrane presents a significant obstacle for a large size glycopeptide such as vancomycin to enter into their cells [3].

We have developed a method to increase salt resistance, serum proteolytic stability, and LPS neutralizing activities of antimicrobial peptides by adding bulky non-nature amino acids to their termini [4,5]. The two terminal β-naphthylalanine residues of S1-Nal-Nal (Ac-KKWRKWLAKKNalNal-NH_2_) were inserted into the hydrophobic lipid A motif of LPS micelles [6]. The extra hydrophobic interactions of S1-Nal-Nal then resulted in greater membrane permeabilization and translocation of the peptide into the membrane. Similarly, derivatives with phenylalanine-(Phe-P-113), β-naphthylalanine- (Nal-P-113), β-diphenylalanine-(Dip-P-113), and β-(4,4’-biphenyl)alanine-(Bip-P-113) substituted histidine-rich antimicrobial peptide P-113 (Ac-AKRHHGYKRKFH-NH_2_) were developed [6,7]. Among these derivatives, Bip-P-113 displayed enhanced salt resistance, serum proteolytic stability, peptide-induced permeabilization, zeta potential measurements, LPS aggregation, and in vitro and in vivo LPS neutralizing activities [8].

Polymyxin B and its derivatives were found to interact with anionic LPS located in the outer leaflet of the outer membrane (OM) of Gram-negative bacteria. The derivatives of polymyxin B were later found to function as “permeabilizers” or “potentiators” that sensitize bacteria to other antibiotics or potentiate the action of other antibiotics [9]. Moreover, synergistic effect between colistin and bacteriocins was found which led to the control of Gram-negative bacteria and reduction of antibiotic toxicity [10]. Recently, ionic silver (Ag^+^) in silver nitrate salt (AgNO_3_) was found to increase Gram-negative bacteria outer membrane permeability and to render vancomycin active against Gram-negative bacteria [11]. Synergistic effects were also found between highly membrane-active antimicrobial peptides and antibiotics with intracellular targets [12]. However, less is known about the synergism between antimicrobial peptides and vancomycin. Based on the strong peptide-induced permeabilization and interactions between P-113 derivatives and POPC:LPS (1-palmitoyl-2-oleoylphosphatidylcholine: lipopolysaccharides) large unilamellar vesicles (LUVs) studied [8], we hypothesized that these peptides may act as “permeabilizers” or “potentiators” to sensitize Gram-negative bacteria such as *E. coli* against vancomycin and to boost synergistic effects with vancomycin against Gram-positive bacterial persister cells.

The aim of this study was to investigate the antibacterial activity of P-113 and its derivatives (Table 1) and to assess their ability to be used in combination with vancomycin against *E. coli* as well as vancomycin-resistant Gram-positive bacteria. The molecular mechanisms of the synergistic effect between P-113 and its derivatives and vancomycin were also studied.

## 2. Results

### 2.1. Antimicrobial Activity

We first explored the antibacterial activity of vancomycin, P-113, and P-113 derivatives against vancomycin-resistant *Enterococcus faecium* (VRE), vancomycin-intermediate *S. aureus 01* (VISA 01), vancomycin-intermediate *S. aureus 02* (VISA 02), vancomycin-intermediate *S. aureus 03* (VISA 03), vancomycin-resistant *S. aureus 01* (VRSA 01), vancomycin-resistant *S. aureus 02* (VRSA 02), and vancomycin-resistant *S. aureus 03* (VRSA 03). As expected, vancomycin had diminished or no activities against these VRE (64 µg/mL), VISA (4 µg/mL), VRSA (32 µg/mL), and *E. coli* (>64 µg/mL) strains (Table 2). On the other hand, all of the peptides with bulky non-nature amino acids demonstrated promising activities with minimal inhibitory concentrations (MICs) ranging from 4–64 µg/mL against these bacterial strains (Table 2). P-113 and Phe-P-113 only had limited antibacterial activities (MICs > 64 µg/mL) [8]. Among these peptides, Bip-P-113 and Nal-P-113 demonstrated the best MICs (Table 2).

### 2.2. Bacterial Membrane Permeabilization

To determine the mechanism of antibacterial action of the antimicrobial peptides, we evaluated the ability of peptides to permeabilize intact VRE (Gram-positive bacteria) and *E. coli* (Gram-negative bacteria) membranes by measuring the peptide-induced leakage of the fluorescent dye calcein from bacterial cells (Figure 1A,B). All peptides showed substantial release of calcein from these two bacterial cells. In general, peptides with bulky non-nature amino acid substitutions demonstrated better bacterial membrane permeabilization ability.

### 2.3. Bacterial Killing Kinetics

After determining the MICs and bacterial membrane permeabilities, we then assessed the killing kinetics of the peptides against the VRE (Gram-positive) and *E. coli* (Gram-negative) strains. As can be seen from Figure 2A, P-113 derivatives with bulky non-nature amino acid substitutions demonstrated fast bactericidal activity and were capable of eliminating a high starting inoculum of the VRE strain (5 × 10^5^ CFU/mL) even at 1 × MIC. The rapid bactericidal activity of Bip-P-113, Nal-P-113, and Dip-P-113 may be due to their ability to rapidly permeabilize the bacterial membrane as described in Figure 1. In contrast to the peptides with non-nature amino acid substitutions, P-113, Phe-P-113, and vancomycin showed slower or limited killing kinetics at 1 ×MIC (Figure 2A). It is evident in the killing kinetics studies that Nal-P-113 had the fastest bactericidal activity than Bip-P-113, and then followed by Dip-P-113 in *E. coli* (Figure 2B). These results indicate that Nal-P-113 and Bip-P-113 are more capable of permeabilizing *E. coli* membrane than the other peptides. Again, P-113, Phe-P-113, and vancomycin showed limited bactericidal activity.

### 2.4. Synergistic Effect with Vancomycin in the Presence of a Sub-Inhibitory Concentration (¼ ×MIC) of Peptides

Due to the strong antimicrobial activity, bacterial membrane permeability, and killing kinetics, we hypothesized that these peptides could be used to resensitize vancomycin-resistant Gram-positive bacterial strains against vancomycin and even to have antibacterial activities against Gram-negative bacterial strains such as *E. coli*. To assess this goal, synergetic activities of P-113 and its derivatives at a sub-inhibitory concentration (¼ ×MIC) with vancomycin were determined by the checkerboard assay as described previously (Table 3). The fractional inhibitory concentration indices (FICI) were interpreted as follows: ≤0.5, “synergy”; >0.5–4, “no interaction”; and >4, “antagonism” [13]. Both P-113 and Phe-P-113 showed no synergy while combined with vancomycin. Dip-P-113 had only synergistic effects against VRE, VRSA 02, and *E. coli*. Nal-P-113 had synergistic effects against VRE, VISA 03, VRSA 01, VRSA 02, VRSA 03, and *E. coli*. Bip-P-113 demonstrated the best synergy with vancomycin against all of the bacterial strains tested.

### 2.5. Mechanism of Resensitization of Bacteria to Vancomycin by P-113 and its Derivatives

We then used boron-dipyrro-methene (BODIPY) labeled vancomycin to study the possible mechanism of synergism between the designed peptides, vancomycin, and the bacterial strains [8]. The results demonstrated that P-113 derivatives with bulky non-nature amino acids markedly enhanced the entry of BODIPY-labeled vancomycin into the VRE strain studied with the order Bip-P-113 = Nal-P-113 > Dip-P-113 > Phe-P-113 = P-113 (Figure 3). Similarly, the order of the BODIPY-labeled vancomycin entering into the Gram-negative *E. coli* strain is Bip-P-113 > Nal-P-113 > Dip-P-113 > Phe-P-113 = P-113 (Figure 4).

### 2.6. P-113 and its Derivatives Attenuate Vancomycin-Induced LPS Release

Antibiotic-induced release of LPS from Gram-negative bacteria has been shown to be associated with the deterioration of the patients [14,15]. Previously, the antimicrobial peptide CLP-19 was shown to reduce antibiotic-induced release of LPS from Gram-negative bacteria [14]. CLP-19 was derived from the core domain of Limulus anti-LPS factor (LALF; amino acids 31–52) with a sequence of (CHYRIKPTFRRLKWKYKGKFWC) and a head to tail disulfide bond. It is likely that the positive charges of CLP-19 can interact with the negative charges of LPS and the hydrophobic part of CLP-19 can bind to the fatty acyl groups of lipid A. The electrostatic attractions and hydrophobic interactions may contribute to the binding and clearance of antibiotic-induced liberation of LPS. In the present study, we measured the concentration of LPS in supernatants and investigated the effect of P-113 and its derivatives on vancomycin-induced release of LPS by limulus amebocyte lysate (LAL) assay (Figure 5) [16,17,18,19]. When *E. coli* cells were treated with the bulky non-nature amino acids substituted P-113 derivatives or the combination of vancomycin and P-113 derivatives, the concentration of LPS in the supernatants decreased tremendously compared to the PBS-treated control cultures (Figure 5).

## 3. Discussion

Vancomycin was used to eradicate bacterial cells by inhibiting peptidoglycan biosynthesis. Gram-negative bacteria use LPS, the major component of the outer membrane, as a protective shield to prevent large glycopeptide antibiotics such as vancomycin from being transported to intracellular targets. Antimicrobial peptides have been shown to act synergistically with vancomycin against vancomycin-persistent Gram-positive bacterial cells [13]. Corbett et al. reported a study using a polymyxin B derivative (SPR741) to potentiate the efficacy of existing antibiotics whose spectrum of activity is limited due to the permeability barrier of the outer membrane of Gram-negative bacteria [9]. MICs were reduced 32–8000-fold against *E. coli* and *Klebsiella pneumoniae* in 8 out of 35 antibiotics while combined with SPR741. However, vancomycin is not listed in these 35 antibiotics.

Recently, it was found that *E. coli* is susceptible to vancomycin through cold stress [3]. Moreover, the mechanism of action of vancomycin to eradicate *E. coli* cells was through inhibition of peptidoglycan biosynthesis, which is similar to the mechanism of action of vancomycin to Gram-positive bacteria [3]. It was also shown that silver ion can increase membrane permeability of Gram-negative bacteria and can potentiate the Gram-positive-specific antibiotic vancomycin against Gram-negative bacteria [11]. These results all indicated that antimicrobial peptides with strong membrane permeability could be used to strengthen our arsenal by using existing Gram-positive-specific antibiotics such as vancomycin against Gram-negative bacterial infections.

Dathe and coauthors used POPC:LPS as model prokaryotic membranes to study peptide activity against *E. coli* and the role of LPS [20]. Previously, we have evaluated the membrane-permeability of P-113 and its derivatives by measuring the release of calcein from various phospholipid vesicles such as POPC:POPG (3:1, mol/mol) LUVs (mimic the anionic bacterial membrane) and POPC:LPS (4:1, mol/mol) LUVs (served as surrogates of Gram-negative bacterial membranes that contain LPS) [8]. P-113 and Phe-P-113 displayed only weak abilities in causing calcein leakage from POPC:POPG and POPC:LPS LUVs. Nal-P-113, Dip-P-113, and Bip-P-113 were shown to possess dose-dependent calcein leakage on POPC:POPG and POPC:LPS LUVs. Among these peptides, Bip-P-113 demonstrated the strongest ability to induce calcein leakage. The results of calcein leakages indicated that the membrane-permeability of the peptides were concordant with the length, and not the width of the bulky non-nature amino acids. Similar results were observed in the present study. Bip-P-113, with the lengthy side chain amino acids, showed the best ability to disturb VRE and *E. coli* outer membrane and enhance the entry of vancomycin into these bacteria. Bip-P-113 also demonstrated the best synergy with vancomycin against the Gram-positive VRE, VISA 01, VISA 02, VISA 03, VRSA 01, VRSA 02, and VRSA 03, and the Gram-negative *E. coli*. In addition to the above-mentioned synergistic effects, the toxicity of the peptides was also determined by measuring cell death using MTT ((3-(4,5-dimethyl-2-thiazolyl)-2,5-diphenyl-2H-tetrazolium bromide) assays against human fibroblasts (HFW) and by measuring hemolysis of human red blood cells (hRBCs) [8]. The results indicated that at higher concentrations (>25 µg/mL), peptides with bulky hydrophobic side chains (Nal-P-113, Bip-P-113, and Dip-P-113) caused higher cell toxicity (10–20% cell death and hemolysis) compared to that of controls. However, in an in vivo endotoxemia mouse model studies, we did not observe any adverse effect in mice treated with 10 mg/kg of Bip-P-113 (0.2 mg/mouse) [8].

Antimicrobial peptides have demonstrated LPS neutralization, wound healing, and anticancer activity [21]. Antimicrobial peptide CLP-19 was shown to effectively reduce the antibiotic-induced release of LPS through neutralization of LPS [14]. Previously, we have demonstrated that the non-nature bulky amino acid substituted P-113 possess LPS binding and neutralizing activity through extra hydrophobic interactions with the lipid A motif of LPS [8]. Similar to the CLP-19 study, all of the P-113 derivatives were found to greatly attenuate vancomycin induced LPS release.

Recently, fluorescent labeled antibiotics were used to characterize the mechanism of bacteria resistance, create new methods to screen and design novel antibiotics. Herein, we have used BODIPY-labeled vancomycin to study the structure-activity relationship of P-113 and its derivatives in affecting the uptake of vancomycin in VRE and *E. coli* strains (Figure 6). Our results indicated that the peptide-incurred uptake of vancomycin were concordant with the length, and not the width of the bulky non-nature amino acids. Similar relationships were also observed in the synergism with vancomycin and attenuation of vancomycin induced LPS release. In addition to the above-mentioned findings, we have found that these antimicrobial peptide-induced fluorescent intensity changes of BODIPY-labeled vancomycin inside bacterial cells may be used as a fast and large scale method to screen antimicrobial peptides’ synergism with vancomycin against Gram-negative and Gram-positive bacteria. Such a study is in progress in our laboratory.

## 4. Materials and Methods

### 4.1. Materials

All peptides were synthesized from Kelowna International Scientific Inc. (Taipei, Taiwan). The identity of the peptides was checked by electrospray mass spectroscopy and the purity (>95%) was assessed by high-performance liquid chromatography (HPLC). Vancomycin hydrochloride was purchased from Bio Basic Inc. (Toronto, ON, Canada). BODIPY-labeled Vancomycin powder was purchased from Thermo Fisher Scientific (Waltham, MA, USA). Mueller-Hinton broth (MHB) and Tryptic Soy broth (TSB) were purchased from Becton, Dickinson and Company (Franklin Lakes, NJ, USA). Agar was purchased from Condalab (Torrejón de Ardoz, Madrid, Spain). Calcein-AM was purchased from Sigma-Aldrich (St. Louis, MO, USA).

### 4.2. Bacterial Strains and Culture Conditions

Clinical isolated multidrug-resistant strains were provided from Tsai-Ling Yang, National Health Research Institutes (Miaoli, Taiwan), including vancomycin-intermediate *Staphylococcus aureus* 01, 02, and 03 (abbreviated as VISA 01, VISA 02, and VISA 03). *Escherichia coli* (ATCC 25922) and *Enterococcus faecium* (BCRC 15B0132, VRE) were purchased from Bioresources Collection & Research Center (BCRC, FIRDI, Hsinchu, Taiwan).

Bacteria were cultured in sterilized MHB or TSB at 37 °C with 150 rpm shaking overnight. The turbidity of bacteria was determined by measuring absorbance of optical density at 600 nm (OD_600_ = 1, equal to approximately 10^8^ CFU/mL) with UV/Visible spectrophotometer (Biochrom, Cambridge, UK).

### 4.3. Serial Passage

For the purpose of inducing vancomycin-resistant *Staphylococcus aureus* (VRSA), the VISA01, VISA02, and VISA03 bacteria were incubated overnight in culture medium, and transferring 50 µL of bacteria to 5 mL of fresh medium containing 2 μg/mL of vancomycin then incubated overnight. The bacteria were transferred to new tube with doubling vancomycin concentration and then incubated overnight again. The concentration of vancomycin during serial passage was 2, 4, 8, 16, and 32 µg/mL. The procedure was repeated until meeting the final concentration of 32 µg/mL. After serial passage, the cell resistance was confirmed by using an MIC assay.

### 4.4. Antimicrobial Activity Assay

The antibacterial activities of peptides were determined by the standard broth microdilution method according to the guidelines of the Clinical and Laboratory Standards Institute (CLSI). Briefly, *E. coli* ATCC 25922 were incubated in MHB, and *E. faecium* BCRC 15B0132, VISA and VRSA strains were incubated in TSB overnight at 37 °C. The cell cultures were regrown to the mid-log phase and subsequently diluted to a final concentration of 5 × 10^5^ CFU/mL. The 1 µL peptides or antibiotic were loaded into each well at the final concentration of 64, 32, 16, 8, 4, 2, 1, and 0.5 µg/mL, then loaded 99 µL diluted microbes into each well of polypropylene 96-well plate. After 16 h of incubation at 37 °C, the turbidity at OD_600_ was measured by the microplate reader. The minimal inhibitory concentration (MIC) value was defined as the lowest concentration of peptide or antibiotic, which inhibited the 90% visible growth of bacteria. All experiments were repeated three times independently.

### 4.5. Checkerboard Assay

The synergistic effects of peptides in combination with vancomycin were investigated by checkerboard assay with slight modification. The peptides were fixed concentration at ¼ × MIC, then in combination with antibiotics at various concentrations in each well of 96-well plate. Then the peptide/antibiotic were mixed with inoculum (5 × 10^5^ CFU/mL) and incubated at 37 °C for 16 h. The fractional inhibitory concentration (FIC) index was calculated as follows: FIC index = (MIC of drug A in combination/MIC of drug A alone) + (MIC of drug B in combination/MIC of drug B alone). An FIC index of ≤ 0.5 was defined as synergy, indifference was defined as an FIC index of 0.5 < FICI ≤ 4, antagonism was defined as an FIC index of > 4. The experiments were performed in triplicate.

### 4.6. Confocal Laser Scanning Microscopy

The fluorescence experiments were performed with slight modification as described previously [22]. Briefly, the bacteria (10^7^ CFU/mL) were incubated with 0.5 ×MIC antimicrobial peptides for 30 min and then treated with BODIPY-labeled vancomycin (2 μg/mL) for 30 min. Samples treated with BODIPY-labeled vancomycin only served as a control. After incubation, bacteria were centrifuged at 8000× *g* for 5 min, washed three times with PBS to remove unbinding fluorescence dye, and resuspended by PBS. The resuspended solution were loaded on the glass slides (Polysine™, Thermo Fisher Scientific, Waltham, MA, USA) and visualized under confocal laser scanning microscope (LSM 510 META, Carl Zeiss, Jena, Thüringen, Germany) equipped with a 64× oil objective lens (Carl Zeiss, Jena, Thüringen, Germany).

### 4.7. Calcein Leakage Assay

Membrane permeabilization of bacteria by peptides was investigated by measuring the leakage of the preloaded fluorescent dye, calcein. Calcein acetoxymethyl ester (calcein-AM) is a nonfluorescent dye and can diffuse across cell membranes to load into the bacteria. After entering the bacterial cells, the calcein-AM is hydrolyzed by cytoplasmic esterases, and yielding the fluorescent derivative calcein (C_30_H_26_N_2_O_13_, molecular weight of 623 g/mol), which has excitation and emission wavelengths of 485 nm and 510 nm, respectively. The bacteria were grown to mid-log phase at 37 °C, and then harvested by centrifugation, washed twice with PBS, and then adjusted to OD600 of 1.0 in PBS containing 10% (vol/vol) MHB or TSB. Then the bacteria were incubated with 3 μM calcein-AM for 1 h at 37 °C. Calcein-AM loaded cells were harvested by centrifugation (3000 × g, 10 min), and resuspended to 10^7^ CFU/mL by PBS. Aliquots of 100 μL were then added into a sterile black-wall 96-well plate, then treated with 0.5 × MIC of peptides in each well and the calcein leakage was measured immediately. Bacteria treated with sterile water were served as negative controls. Calcein leakage experiment was measured for 60 min by using a fluorescence plate reader (VICTOR3, PerkinElmer, USA) equipped with a 485 nm excitation filter and a 510 nm emission filter. Cobalt (Co^2+^, chloride salt, 2 μM) was used to quench the fluorescence of calcein released into the extracellular environment. Membrane permeabilization (%) was calculated as the absolute percent of calcein leakage by peptides with respect to calcein-AM loaded with no-peptide treated cells.

### 4.8. Time Killing Assay

The bactericidal activities of vancomycin, P-113, Phe-P-113, Bip-P-113, Dip-P-113, and Nal-P-113 against *E. faecium* BCRC 15B0132 (VRE) and *E. coli* ATCC 25922 were assessed by the time-kill analysis. The cells were grown overnight in TSB or MHB at 37 °C with shaking, and then fresh TSB or MHB was inoculated for growth to mid-logarithmic phase. Bacteria (5 × 10^5^ CFU/mL) were treated with 1 × MIC vancomycin, P-113, Phe-P-113, Bip-P-113, Dip-P-113, and Nal-P-113 for 0.5, 1, 2, 4, 6, 10, and 24 h at 37 °C. Bacteria without peptides and vancomycin served as a control. Aliquots were removed at each time points, and 10-fold serial dilutions of the aliquot were plated onto the Müller-Hinton agar. The numbers of CFU were counted after 24 h of incubation at 37 °C. The detection limit for the colony counts was 2log_10_ CFU/mL.

### 4.9. Anti-Endotoxin Studies

*E. coli* ATCC 25922 were incubated at the mid-log phase (10^4^ CFU/mL) and treated with peptide alone (at 1 × MIC) or combination with vancomycin (both at 0.5 × MIC) at 37 °C for 6 h. The samples were filtered through a pyrogen-free 0.2 µm pore filter (Acrodisc, Pall Corporation, Port Washington, NY, USA) and the endotoxin level was detected by limulus amebocyte lysate (LAL) PYROCHROME^®^ test (Associates of Cape Cod, East Falmouth, MA, USA). The kinetic turbidity was measured using the microplate reader (SpectraMax ABS, Molecular Devices, San Jose, CA, USA).

### 4.10. Statistical Analysis

All statistical results are expressed as the mean ± SEM and were analyzed using a one-way ANOVA (analysis of variance). Statistical analysis was performed using GraphPad Prism version 5.0 (San Diego, CA, USA), where *p* < 0.05 was considered to indicate a statistically significant difference.

## 5. Conclusions

In conclusion, our results demonstrated that the bulky non-nature amino acid substituted P-113 derivatives display compelling antibacterial and synergistic activities with vancomycin against vancomycin-resistant *E. faecium*, *S. aureus*, and wild-type *E. coli*. Moreover, these antimicrobial peptides also reduced antibiotic-induced releasing of LPS from Gram-negative bacteria. Among these peptides, Bip-P-113 demonstrated the best MICs, antibiotics synergism, and supernatant LPS neutralizing activities against the bacteria studied. Our results should be useful in the development of new antimicrobial peptides and peptidomimetics with both antimicrobial activity and antibiotics synergism for potential therapeutic applications.

## Figures and Tables

**Figure 1 ijms-21-04578-f001:**
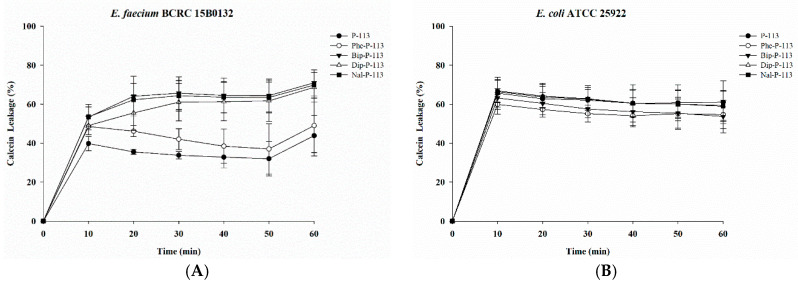
Membrane permeabilization of vancomycin-resistant *Enterococcus faecium* (VRE) and *Escherichia coli* strains. The membrane permeabilization of (**A**) *E. faecium* BCRC 15B0132, and (**B**) *E. coli* ATCC 25922 by P-113, Phe-P-113, Bip-P-113, Dip-P-113 and Nal-P-113 for 60 min exposure. Calcein-AM loaded cells (10^7^ CFU/mL) were resuspended by PBS, and the aliquots of 100 μL were added into a sterile black-wall 96-well plate, then treated with 0.5 × MIC of peptides in each well and measured the calcein leakage immediately. The experiments were performed in triplicate.

**Figure 2 ijms-21-04578-f002:**
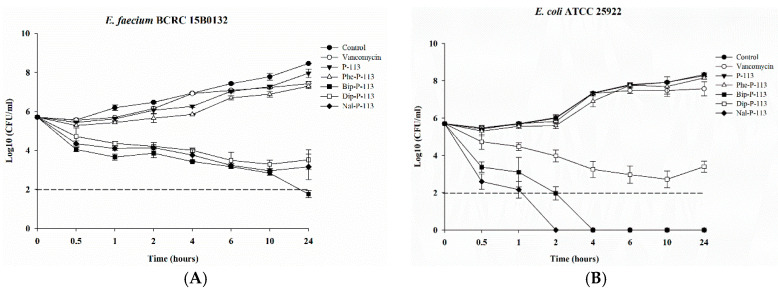
Killing kinetics of vancomycin, P-113 and its derivatives against VRE and *E. coli* strains. 5 × 10^5^ CFU/mL of (**A**) *E. faecium* BCRC 15B0132, and (**B**) *E. coli* ATCC 25922 were treated with 1 × MIC vancomycin, P-113, Phe-P-113, Bip-P-113, Dip-P-113, and Nal-P-113 for 0.5, 1, 2, 4, 6, 10, and 24 h at 37 °C. Bacteria without peptides and vancomycin were used as a control. Aliquots were removed at each time points, and 10-fold serial dilutions of the aliquot were plated onto Müller-Hinton agar. The numbers of CFU were counted after 24-hours incubation at 37 °C. The detection limit for the colony counts was 2log_10_ CFU/mL. The experiments were performed in triplicate.

**Figure 3 ijms-21-04578-f003:**
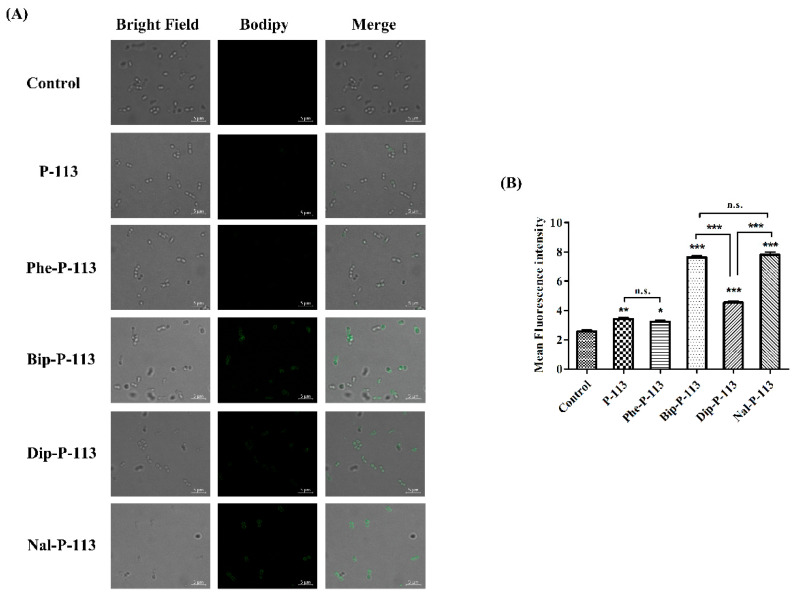
P-113 and its derivatives increase the uptake of BODIPY-labeled vancomycin in *E. faecium*. (**A**) Fluorescence images of 10^7^ CFU/mL *E. faecium* BCRC 15B0132 treated with 0.5 ×MIC of P-113, Phe-P-113, Bip-P-113, Dip-P-113, and Nal-P-113 at 37 °C for 30 min, then treated BODIPY-labeled vancomycin for 30 min (scale bar represents 5 µm). (**B**) Mean fluorescence intensity of P-113, Phe-P-113, Bip-P-113, Dip-P-113, and Nal-P-113 treated in *E. faecium* BCRC 15B0132. Samples treated with BODIPY-labeled vancomycin only served as a control. Results are presented as means ± SEM, * *p* < 0.05, ** *p* < 0.01, *** *p* < 0.001 compared with control. n.s. = no significant differences. The experiments were performed in triplicate.

**Figure 4 ijms-21-04578-f004:**
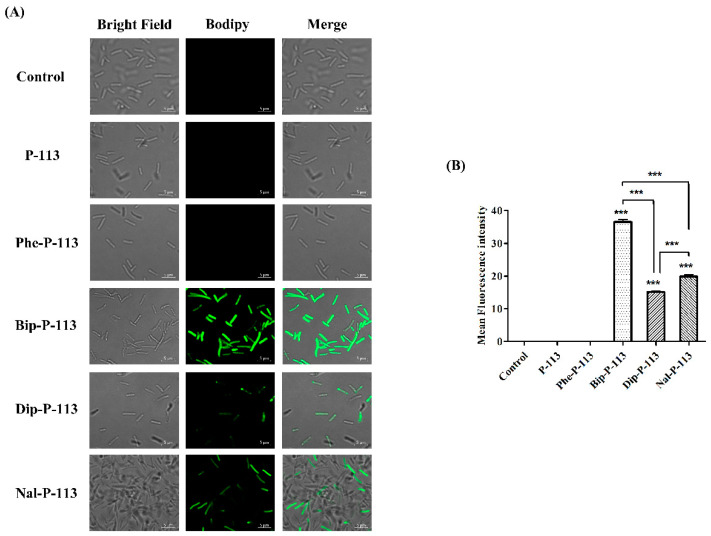
P-113 and its derivatives increase the uptake of BODIPY-labeled vancomycin in *E. coli*. (**A**) Fluorescence images of 10^7^ CFU/mL *E. coli* ATCC 25922 treated with 0.5 ×MIC of P-113, Phe-P-113, Bip-P-113, Dip-P-113, and Nal-P-113 at 37 °C for 30 min, then treated with BODIPY-labeled vancomycin for 30 min (scale bar represents 5 µm). (**B**) Mean fluorescence intensity of P-113, Phe-P-113, Bip-P-113, Dip-P-113, and Nal-P-113 treated in *E. coli* ATCC 25922. Samples treated with BODIPY-labeled vancomycin only served as a control. Results are presented as means ± SEM, *** *p* < 0.001 compared with control. The experiments were performed in triplicate.

**Figure 5 ijms-21-04578-f005:**
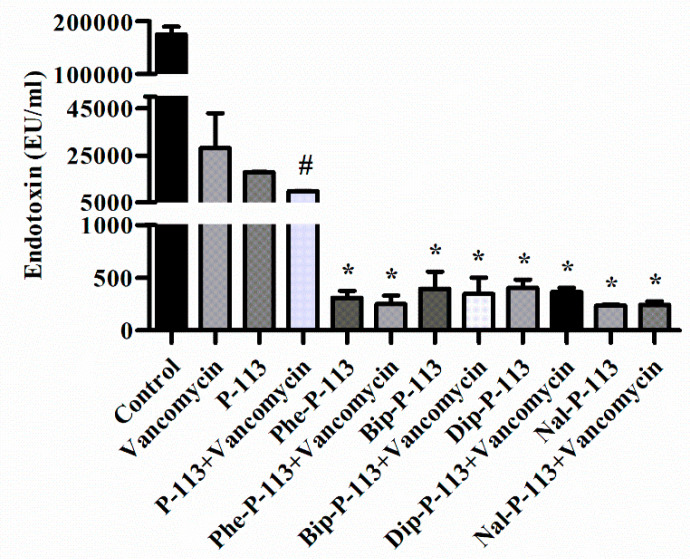
The influence of vancomycin, P-113 and its derivatives on LPS release from *E. coli*. *E. coli* ATCC 25922 were incubated at the mid-log phase (10^4^ CFU/mL) and treated with peptide alone (at 1 × MIC) or combination with vancomycin (both at 0.5 × MIC) at 37 °C for 6 h. The samples were filtered through 0.2 µm pore filter and the endotoxin level was detected by LAL assay. # *p* < 0.05 compared with vancomycin only, * *p* < 0.001 compared with vancomycin only. The experiments were performed in triplicate.

**Figure 6 ijms-21-04578-f006:**
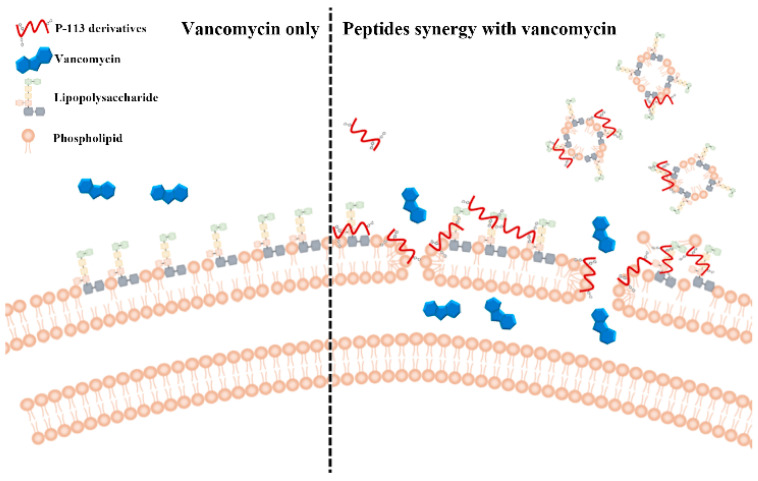
Synergistic mechanism of action proposed for P-113 derivatives and vancomycin. For vancomycin, the efficacy was limited due to the permeability barrier of the outer membrane of Gram-negative bacteria (left panel). However, P-113 derivatives (e.g., Bip-P-113, Dip-P-113, and Nal-P-113) could disturb the outer membrane of Gram-negative bacteria and enhance the entry of vancomycin into these bacteria (right panel). Moreover, P-113 derivatives could bind and neutralize LPS through extra hydrophobic interactions with the lipid A motif of LPS (right panel).

**Table 1 ijms-21-04578-t001:** Sequences of P-113, Phe-P-113, Bip-P-113, Dip-P-113 and Nal-P-113.

Name	Chemical Structure ^a^	Sequence ^b^	Molecular Weight (Da)
P-113	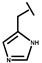	Ac- A K R His His G Y K R K F His -NH_2_	1605.86
Phe-P-113	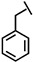	Ac- A K R Phe Phe G Y K R K F Phe -NH_2_	1635.99
Bip-P-113	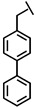	Ac- A K R Bip Bip G Y K R K F Bip -NH_2_	1864.97
Dip-P-113	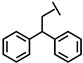	Ac- A K R Dip Dip G Y K R K F Dip -NH_2_	1864.06
Nal-P-113	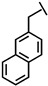	Ac- A K R Nal Nal G Y K R K F Nal -NH_2_	1786.03

^a^ Side chain structure of amino acids at positions 4, 5, 12. ^b^ Bip: β-(4.4’-biphenyl)alanine; Dip: β-diphenylalanine; Nal: β-Naphthylalanine.

**Table 2 ijms-21-04578-t002:** MICs of Vancomycin, P-113, Phe-P-113, Bip-P-113, Dip-P-113 and Nal-P-113. The experiments were performed in triplicate.

Bacterial Strains	MIC (μg/mL)					
	Vancomycin	P-113	Phe- P-113	Bip- P-113	Dip- P-113	Nal- P-113
*E. faecium* BCRC 15B0132 (VRE)	64	>64	>64	4	4	4
Vancomycin-intermediate *S. aureus* 01 (VISA01)	4	>64	>64	16	16	8
Vancomycin-intermediate *S. aureus* 02 (VISA02)	4	>64	>64	16	16	8
Vancomycin-intermediate *S. aureus* 03 (VISA03)	4	>64	>64	8	16	8
Vancomycin-resistant *S. aureus* 01 (VRSA01)	32	>64	>64	16	32	8
Vancomycin-resistant *S. aureus* 02 (VRSA02)	32	>64	>64	16	32	8
Vancomycin-resistant *S. aureus* 03 (VRSA03)	32	>64	>64	8	32	16
*E. coli* ATCC 25922	>64	>64	>64	32	64	32

**Table 3 ijms-21-04578-t003:** Synergistic effects of P-113, Phe-P-113, Bip-P-113, Dip-P-113, and Nal-P-113 in combination with vancomycin against bacterial strains studied. All of the experiments were performed in triplicate.

Strains	AMP (μg/mL) (¼ ×MIC)									
	P-113		Phe-P-113		Bip-P-113		Dip-P-113		Nal-P-113	
	VAN ^a^	FICI ^b^	VAN	FICI	VAN	FICI	VAN	FICI	VAN	FICI
VRE	32	0.75	32	0.75	8	0.38	16	0.50	16	0.50
VISA 01	4	1.25	4	1.25	1	0.50	4	1.25	2	0.75
VISA 02	4	1.25	4	1.25	1	0.50	4	1.25	2	0.75
VISA 03	4	1.25	4	1.25	1	0.50	2	0.75	1	0.50
VRSA 01	32	1.25	32	1.25	2	0.31	16	0.75	8	0.50
VRSA 02	32	1.25	32	1.25	2	0.31	8	0.50	8	0.50
VRSA 03	32	1.25	32	1.25	4	0.38	16	0.75	4	0.38
*E. coli* ATCC 25922	>64	1.25	64	0.75	16	0.38	32	0.50	16	0.38

^a^ VAN, vancomycin (μg/mL). ^b^ FICI, fractional inhibitory concentration index, FICI ≤ 0.5, synergy; 0.5 < FICI ≤ 4, no interaction; FICI > 4, antagonism [13].

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
