# Peer review of "Antimicrobial Peptides Display Strong Synergy with Vancomycin Against Vancomycin-Resistant E. faecium, S. aureus, and Wild-Type E. coli"

_ijms, 2020, doi:10.3390/ijms21134578_

Round 1

Reviewer 1 Report

In “Antimicrobial peptides display strong synergy with vancomycin against vancomycin-resistant E. faecium, S. aureus, and wild-type E. coli” the authors describe the synergistic interactions between peptide derivatives and the antibiotic vancomycin. The study is an extension of a previous publication in which similar studies were carried out with the same peptides but different antibiotics, so the rationale of the study was to extend the idea to vancomycin. The methodology is strong and the results describe interactions between the peptide derivatives and the vancomycin at inhibiting Gram(+) and Gram(-) bacteria. This reviewer finds the presentation weak in some parts.

- The introduction will benefit by adding a more descriptive narrative in support of the rationale of the study.

- As the discussion highlights the length and width of the structures, a figure describing the chemical structure of the peptide derivatives should be added, probably combined with table 1.

- Tables are lacking a head with proper description of experimentation.

- Sentence in lines 65-66 sound non-specific and should mention the actual measurement units instead of just “activities”

- line 87 “were capable”

- In legend to figure 2, the timing of the experiments is unclear. All experiments were stopped at 24 h?

- lines 103-104. The sentence should read “re-sensitize vancomycin resistant bacteria against vancomycin”

- In line 115, the title should read “ Mechanism of re-sensitization of BACTERIA TO vancomycin by P-113 and its derivatives”

- In line 134, then treated WITH BODITY-labeled…..”

- In line 141, the antimicrobial peptide CLP-19 needs to be described. How it is different from the present study peptides and how it reduced release of LPS.

- in line 144, mention what is LAL

- In line 187, the sentence needs a reference.

-  In lines 194-194, revise sentence.

- in the Discussion section, address the issue of how these peptides affect membranes of eukaryotic cells and how these effect may impact the peptide’s medical applications.

- The following references are upper case worded and should be corrected: 3, 4, 10, 13

- The following references need abbreviated name of journal: 8, 17

Author Response

Reviewer 1:

In “Antimicrobial peptides display strong synergy with vancomycin against vancomycin-resistant E. faecium, S. aureus, and wild-type E. coli” the authors describe the synergistic interactions between peptide derivatives and the antibiotic vancomycin. The study is an extension of a previous publication in which similar studies were carried out with the same peptides but different antibiotics, so the rationale of the study was to extend the idea to vancomycin. The methodology is strong and the results describe interactions between the peptide derivatives and the vancomycin at inhibiting Gram(+) and Gram(-) bacteria. This reviewer finds the presentation weak in some parts.

1. The introduction will benefit by adding a more descriptive narrative in support of the rationale of the study. 

Response to the reviewer's comment:

We thank the reviewer for the suggestion. We have added the following sentences in the introduction to support the rationale of the study. “Polymyxin B and its derivatives were found to interact with anionic lipopolysaccharide (LPS) located in the outer leaflet of the outer membrane (OM) of Gram-negative bacteria. The derivatives of polymyxin B were later found to function as “permeabilizers” or “potentiators” to sensitize bacteria to other antibiotics or potentiate the action of other antibiotics. Recently, ionic silver (Ag+) in silver nitrate salt (AgNO3) was found to increase Gram-negative bacteria outer membrane permeability and to render vancomycin active against Gram-negative bacteria. Synergistic effects were also found between highly membrane-active antimicrobial peptides and antibiotics with intracellular targets. However, less is known about the synergism between antimicrobial peptides and vancomycin. Based on the strong peptide-induced permeabilization and interactions between P-113 derivatives and POPC:LPS large unilamellar vesicles (LUVs) studied, we hypothesized that these peptides may act as “permeabilizers” or “potentiators” to sensitize Gram-negative bacteria such as E. coli against vancomycin and to boost synergistic effects with vancomycin against Gram-positive bacterial persister cells.”

2. As the discussion highlights the length and width of the structures, a figure describing the chemical structure of the peptide derivatives should be added, probably combined with table 1. 

Response to the reviewer's comment:

We thank the reviewer for the suggestion. We have incorporated the chemical structures of the bulky non-nature amino acids into Table 1 as suggested.

3. Tables are lacking a head with proper description of experimentation.

Response to the reviewer's comment:

We thank the reviewer for the suggestion. We have incorporated the head with proper description of experimentation into the tables.

4. Sentence in lines 65-66 sound non-specific and should mention the actual measurement units instead of just “activities”. 

Response to the reviewer's comment:

We thank the reviewer for the suggestion. We have incorporated the actual measurement units into lines 65-66 as suggested.

5. line 87 “were capable”

Response to the reviewer's comment:

We have revised line 87 to “were capable” as suggested by the reviewer.

6. In legend to figure 2, the timing of the experiments is unclear. All experiments were stopped at 24 h? 

Response to the reviewer's comment:

We thank the reviewer for pointing out this to us. To make it clear, we have implemented the description “Aliquots were removed at each time point, and 10-fold serial dilution of the aliquots were plated onto MH agar” into the legend of Figure 2. All experiments were stopped at 24 h.

7. lines 103-104. The sentence should read “re-sensitize vancomycin resistant bacteria against vancomycin”

Response to the reviewer's comment:

We thank the reviewer for the suggestions. We have revised lines 103-104 into “Because of the strong antimicrobial activity, bacterial membrane permeability, and killing kinetics, we hypothesized that these peptides could be used to re-sensitize vancomycin-resistant Gram-positive bacterial strains against vancomycin”.

8. In line 115, the title should read “Mechanism of re-sensitization of BACTERIA TO vancomycin by P-113 and its derivatives”. 

Response to the reviewer's comment:

We thank the reviewer for the suggestion. We have revised the title in line 115 to “Mechanism of re-sensitization of bacteria to vancomycin by P-113 and its derivatives”.

9. In line 134, then treated WITH BODITY-labeled…..” 

Response to the reviewer's comment:

We have revised line 134 into “then treated with BODIPY-labeled…..” as suggested.

10. In line 141, the antimicrobial peptide CLP-19 needs to be described. How it is different from the present study peptides and how it reduced release of LPS. 

Response to the reviewer's comment:

We thank the reviewer for the suggestions. CLP-19 was derived from the core domain of Limulus anti-LPS factor (LALF; amino acids 31-52) with a sequence of (CHYRIKPTFRRLKWKYKGKFWC) and a head to tail disulfide bond. It is likely that the positive charges of CLP-19 can interact with the negative charges of LPS and the hydrophobic part of CLP-19 can bind to the fatty acyl groups of lipid A. The electrostatic attractions and hydrophobic interactions then contribute to the binding and clearance of antibiotic-induced liberation of LPS. Previously, we have demonstrated that the non-nature bulky amino acid substituted P-113 possess LPS binding and neutralizing activity through similar electrostatic attractions and hydrophobic interactions found in the CLP-19 study. We have implemented these descriptions into the text.

11. in line 144, mention what is LAL 

Response to the reviewer's comment:

We thank the reviewer for pointing out this to us. We have implemented the full description of limulus amebocyte lysate (LAL) assay into the text.

12. In line 187, the sentence needs a reference. 

Response to the reviewer's comment:

We have added a reference to the sentence (ref. 21) as suggested.

13. In lines 194-194, revise sentence. 

Response to the reviewer's comment:

We thank the reviewer for the suggestion. We have revised line 194 to ”Recently, fluorescent labeled antibiotics were used to characterize the mechanism of bacteria resistance, create new methods to screen and design novel antibiotics”.

14. in the Discussion section, address the issue of how these peptides affect membranes of eukaryotic cells and how these effect may impact the peptide’s medical applications. 

Response to the reviewer's comment:

We thank the reviewer for the suggestions. We have implemented the following discussions regarding how these peptides affect membranes of eukaryotic cells and how these effect may impact the peptide’s medical applications into the text as suggested. “In addition to the above-mentioned synergistic effects, the toxicity of the peptides was determined by measuring cell death using MTT ((3 - (4,5 – dimethyl – 2 - thiazolyl) - 2,5 – diphenyl - 2H - tetrazolium bromide) assays against human fibroblasts (HFW) and by measuring hemolysis of human red blood cells (hRBCs). The results indicated that at higher concentrations (> 25 μg/ml), peptides with bulky hydrophobic side chains (Nal-P-113, Bip-P-113, and Dip-P-113) caused higher cell toxicity (10%–20% cell death and hemolysis) compared to that of controls. However, in an in vivo endotoxemia mouse model studies, we did not observe any adverse effect in mice treated with 10 mg/kg of Bip-P-113 (~0.2 mg/mouse).”

15. The following references are upper case worded and should be corrected: 3, 4, 10, 13. 

Response to the reviewer's comment:

We thank the reviewer for pointing out these to us. We have revised the references.

16. The following references need abbreviated name of journal: 8, 17. 

Response to the reviewer's comment:

We thank the reviewer for pointing out these to us. We have revised the references.

Reviewer 2 Report

This paper by Wu et al is a well written, clear demonstration of the synergic effect of modified antimicrobial peptides with valinomycin on both Gram positive and Gram negative bacteria, therefore I recommend it for publication. However, there are some issues that need to be addressed.

There are some abbreviations that are not explained at their first appearance, like MIC in the abstract, LPS, POPC in the introduction, LAL assay in the results.

In most experiments the number of repetitions is not provided, I found it only (as triplicated) in the checkerboard and antimicrobial activity assay descriptions. In Figure and Table legends this information should be provided. Also, it should be indicated that the Fig 3A and 4A images are representatives of how many experiments.

Author Response

Reviewer 2:

This paper by Wu et al is a well written, clear demonstration of the synergic effect of modified antimicrobial peptides with vancomycin on both Gram positive and Gram negative bacteria, therefore I recommend it for publication. However, there are some issues that need to be addressed.

1. There are some abbreviations that are not explained at their first appearance, like MIC in the abstract, LPS, POPC in the introduction, LAL assay in the results.

Response to the reviewer's comment:

We thank the reviewer for pointing out this to us. We have explained the abbreviations of MIC, LPS, POPC, and LAL in their first appearance.

2. In most experiments the number of repetitions is not provided, I found it only (as triplicated) in the checkerboard and antimicrobial activity assay descriptions. In Figure and Table legends this information should be provided. Also, it should be indicated that the Fig 3A and 4A images are representatives of how many experiments.

Response to the reviewer's comment:

We thank the reviewer for pointing out these to us. All of the experiments were performed in triplicate. We have included the number of repetitions of the experiments in the Figures and Tables as suggested.